# Leveraging functional annotation to identify genes associated with complex diseases

**Wei Liu[1], Mo Li[2], Wenfeng Zhang[2], Geyu Zhou[1], Xing Wu[3], Jiawei Wang[1], Qiongshi Lu[4,5,6], Hongyu Zhao[1,2,7]** *

**1** Program of Computational Biology and Bioinformatics, Yale University, New Haven, CT, United States of America, **2** Department of Biostatistics, Yale School of Public Health, New Haven, CT, United States of America, **3** Department of Molecular, Cellular and Developmental Biology, Yale University, New Haven, CT, United States of America, **4** Department of Biostatistics and Medical Informatics, University of Wisconsin-Madison, WI, United States of America, **5** Department of Statistics, University of Wisconsin-Madison, WI, United States of America, **6** Center for Demography of Health and Aging, University of Wisconsin-Madison, WI, United States of America, **7** Department of Genetics, Yale School of Medicine, New Haven, CT, United States of America

* hongyu.zhao@yale.edu

**Data Availability Statement:** Roadmap epigenomics project data are available at: https://egg2.wustl.edu/roadmap/web_portal/. GTEx gene expression data are available at: https://gtexportal.

## Abstract

To increase statistical power to identify genes associated with complex traits, a number of transcriptome-wide association study (TWAS) methods have been proposed using gene expression as a mediating trait linking genetic variations and diseases. These methods first predict expression levels based on inferred expression quantitative trait loci (eQTLs) and then identify expression-mediated genetic effects on diseases by associating phenotypes with predicted expression levels. The success of these methods critically depends on the identification of eQTLs, which may not be functional in the corresponding tissue, due to linkage disequilibrium (LD) and the correlation of gene expression between tissues. Here, we introduce a new method called T-GEN (**T**ranscriptome-mediated identification of disease-associated **G**enes with **E**pigenetic a**N**notation) to identify disease-associated genes leveraging epigenetic information. Through prioritizing SNPs with tissue-specific epigenetic annotation, T-GEN can better identify SNPs that are both statistically predictive and biologically functional. We found that a significantly higher percentage (an increase of 18.7% to 47.2%) of eQTLs identified by T-GEN are inferred to be functional by ChromHMM and more are deleterious based on their Combined Annotation Dependent Depletion (CADD) scores. Applying T-GEN to 207 complex traits, we were able to identify more trait-associated genes (ranging from 7.7% to 102%) than those from existing methods. Among the identified genes associated with these traits, T-GEN can better identify genes with high (>0.99) pLI scores compared to other methods. When T-GEN was applied to late-onset Alzheimer's disease, we identified 96 genes located at 15 loci, including two novel loci not implicated in previous GWAS. We further replicated 50 genes in an independent GWAS, including one of the two novel loci.

org/home/datasets; GTEx genotype data: v6 dbGaP accession phs000424.v6.p1; v8 dbGaP accession phs000424.v8. GWAS summary stats from LD hub are available at: http://ldsc.broadinstitute.org. All trained models and association test results of T-GEN can be found in https://github.com/vivid-/T-GEN.

**Funding:** Supported in part by NIH grants R01 GM122078, R01 GM134005, and P30 AG021342, and NSF grant DMS 1902903. The funders had no role in study design, data collection and analysis, decision to publish, or preparation of the manuscript.

**Competing interests:** The authors have declared that no competing interests exist.

## Author summary

TWAS-like methods have been widely applied to understand disease etiology using eQTL data and GWAS results. However, it is still challenging to discriminate the true disease-associated genes from those in strong LD with true genes, which is largely due to the mis-identification of eQTLs. Here we introduce a novel statistical method named T-GEN to identify disease-associated genes considering epigenetic information. Compared to current TWAS methods, T-GEN not only identified eQTLs with higher CADD scores and function potentials in gene-expression imputation models, but also identified more disease-associated genes across 207 traits and more genes with high (>0.99) pLI scores. Applying T-GEN in late-onset Alzheimer's disease identified 96 genes at 15 loci with two novel loci. Among 96 identified genes, 50 genes were further replicated in an independent GWAS.

This is a *PLOS Computational Biology* Methods paper.

## Introduction

Genome-Wide Association Studies (GWAS) have been very successful in identifying single nucleotide polymorphisms (SNPs) associated with human diseases [1]. However, most identified SNPs are located in non-coding regions, making it challenging to understand the roles of these SNPs in disease etiology. Several approaches have been developed recently to link genes with identified SNPs and provide insights for downstream analysis [2–6]. PrediXcan [7] and similar methods [8–12] have been developed for utilizing transcriptomic data, such as those from GTEx [13], to interpret identified GWAS non-coding signals and to identify additional disease associated genes. These methods first impute (i.e. predict) gene expression levels from SNP genotypes and then identify disease-associated genes by associating phenotypes with predicted expression levels. At the SNP level, SNPs used in the gene expression imputation models are selected through statistical correlation between these SNPs' genotypes and gene expression levels. Since SNPs in the same LD block are correlated, it is hard to differentiate regulatory SNPs from others statistically, which may lead to incorrect identifications of genes with regulatory SNPs that are in strong LD with true trait/disease genes.

To more accurately identify regulatory eQTLs that play functional roles, we assume that SNPs with active epigenetic annotations are more likely to regulate tissue-specific gene expression [14–18]. As for available tissue-specific epigenetic data, we consider epigenetic marks that are known hallmarks for DNA regions with important functions, such as H3K4me1 signals that are often associated with enhancers [19]. We note that these epigenetic marks have been used to infer regulatory regions and prioritize eQTLs in some published studies [20–23]. Reported GWAS hits are enriched in regions with active epigenetic signals, the Encyclopedia of DNA Elements (ENCODE) project found that 34% of disease-associated SNPs overlap DNA-hypersensitive sites (encompassing 3.9% of the whole genome sequence) [24], and these epigenetic signals can help fine-map true GWAS hits with functional impacts [25]. Based on these previous findings, we use epigenetic signals to select SNPs among all candidate cis-SNPs when modeling the relationship between gene expression levels and SNP genotypes. We have developed a new method, called T-GEN (**T**ranscriptome-mediated identification of disease-associated **G**enes with **E**pigenetic a**N**notation), that leverages tissue-specific epigenetic information to identify disease-associated genes. By prioritizing the SNPs likely having regulatory roles (through the use of epigenetic marks) when building gene expression imputation models,

T-GEN identified SNPs with higher deleterious effects and higher function potential, Further application of T-GEN identified more trait-associated genes in 207 traits, compared to other gene expression imputation models. More specifically, in late-onset Alzheimer's disease (AD), T-GEN identified the largest number of genes with novel loci, indicating the importance of cholesterol transportation, neuron activity and mitochondrial dysfunction in late-onset AD.

## Results

### Method overview

Similar to previous methods [8,26,27], we use two linear models to study gene-level genetic effects on traits mediated by gene expression regulation. Firstly, individual-level genotype and gene expression data are used to build gene expression imputation models for each tissue. One novel feature of our method is the integration of epigenetic data from the Roadmap Epigenomics Project [28] to prioritize regulatory SNPs in gene expression imputation. As a result, SNPs located in regions with active epigenetic marks are more likely to be selected, consistent with the enrichment of epigenetic marks (e.g. H3K4me3 and DNase-I hypersensitivity) in regulatory DNA regions for gene expression [29]. After obtaining tissue-specific gene expression imputation models, we combine them with GWAS summary statistics to identify gene-level associations with disease phenotypes. A schematic workflow of T-GEN is shown in **Fig 1**.

By utilizing a spike-and-slab prior, SNPs regulating gene expression levels were selected considering their epigenetic signals through the following model:

$$Y = X\beta + \epsilon,$$

$$\beta_k \sim \pi_k N(0, \sigma_\beta^2 \sigma^2 I) + (1 - \pi_k)\delta_0,$$

$$logit(\pi_k) = A_k\omega,$$

where $Y$ is the vector of gene expression values in a given tissue for a gene, $X$ is the genotype matrix for the candidate cis-SNPs for the gene, $\beta$ is the tissue-specific effect vector of genotypes on expression level, and $\epsilon$ denotes the random noise. For a cis-SNP $k$, its effect $\beta_k$ follows a mixture prior of a normal distribution and a point mass around 0. The probability $\pi_k$ of the SNP being an eQTL of its nearby gene is linked to its epigenetic annotation matrix $A_k$ via a logit model, with $\omega$ being the annotation coefficient vector. We use a variational Bayesian method [30] to estimate the coefficient vector $\beta$. More details are shown in the Methods section.

For comparison, we also consider four other gene expression imputation methods, including elastic net (elnt) [7,31] used in both PrediXcan and FUSION, linear models with spike-and-slab priors solved by variational Bayes (vb) [32,33], and both elastic net and linear models with spike-and-slab priors applied only to SNPs having active epigenetic signals (elnt.annot and vb.annot). Also, the annotation configuration and incompleteness may also affect the results, which we discussed in S1 Text.

In general, we describe the relationship between the imputed gene expression level and selected SNPs in the form of $\widehat{Y} = X\widehat{\beta}$, where $\widehat{\beta}$ denotes the SNP effect estimates in the imputation models. Different imputation methods lead to different sets of SNPs and effect size estimates. We then use a univariate regression model to test the association between traits and imputed gene expression levels, which was also used for gene-trait testing in many TWAS methods [8,26,27]:

$$T = \mu + \widehat{Y}\kappa_Y + \tau.$$

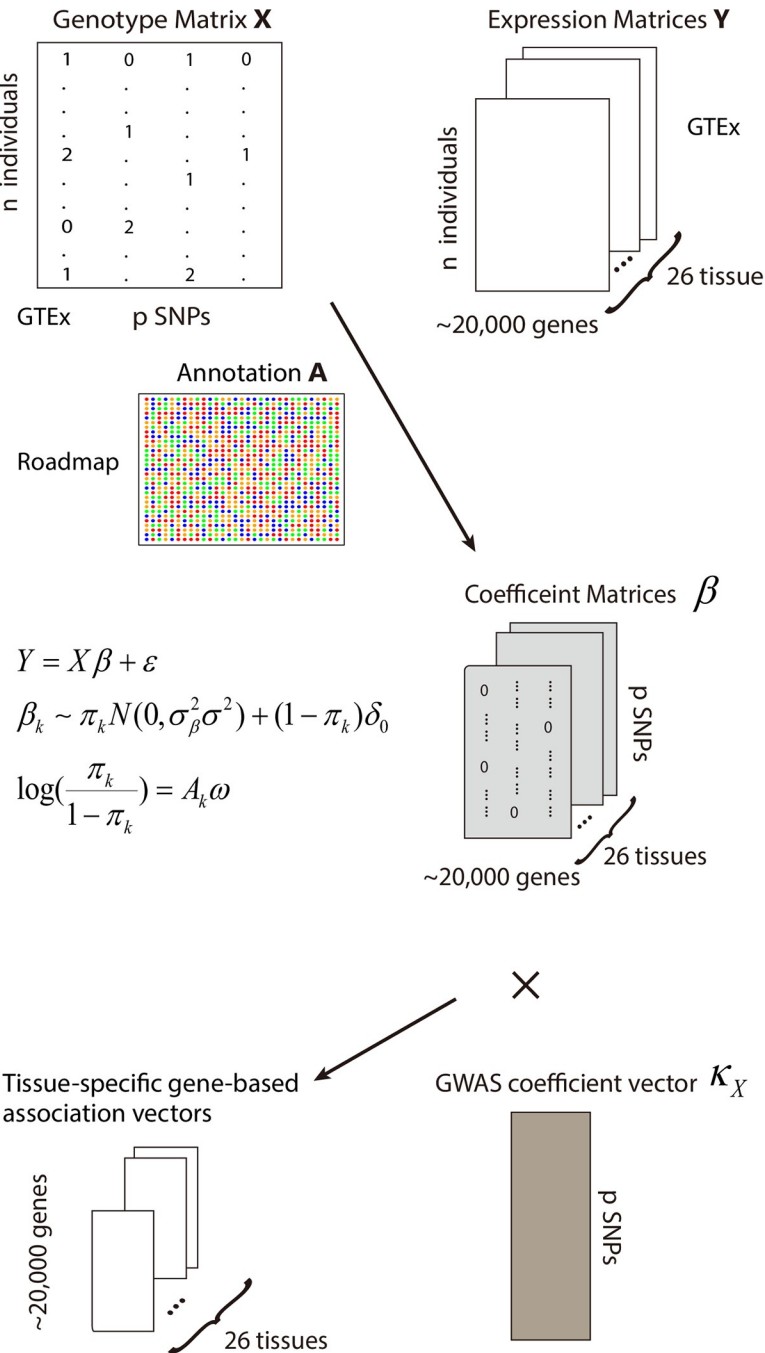

**Fig 1. The general scheme of our method.** Gene expression imputation models were built based on gene expression matrix Y of a specific gene in a tissue, the genotype matrix X and epigenetic annotation matrix A of cis-SNPs of the gene. The annotation matrix was used to select SNPs having regulatory effects on gene expression since we assumed that only part of cis-SNPs have effects on gene expression of the nearby gene. After getting SNP coefficient vectors β in each tissue for the gene, we combined β's with GWAS summary stats and then get the gene-level association statistics for each disease in each tissue.

The z-score of the gene coefficient $\kappa_Y$ is denoted as $z = \frac{\widehat{\kappa_Y}}{se(\kappa_Y)} = \widehat{\beta}^T \Lambda \widehat{Z}_X$, where $\Lambda$ is a diagonal matrix with diagonal elements being the ratio of the standard deviation of SNP genotypes over the standard deviation of the imputed gene expression levels, while $\widehat{Z}_X$ is the z score vector of SNP effects on traits in GWAS. We test the association between each tissue-gene pair and trait. Significant disease-associated tissue-gene pairs are then identified after adjusting for multiple testing. The implemented method and pre-trained models are available on https://github.com/vivid-/T-GEN.

## T-GEN prioritizes biologically functional SNPs in gene expression imputation

To study whether our method can better prioritize functional SNPs in gene expression imputation models, we evaluated the functional states of the identified eQTLs using the ChromHMM-annotated SNP status [34]. T-GEN identified higher percentage eQTLs (47%) having one of the 11 active states (out of 15 states) (**S1 Table**), with an 87% increase over elnt models (25%) and 77% increase over vb (27%) models (**Fig 2A**). Overall, these results demonstrate that our method better identified SNPs with functional potential to regulate gene expression while not increasing the total number of SNPs selected (**S1 Fig**).

We note that T-GEN utilizes the epigenetic information in SNP selections, and the same information is also used in ChromHMM models. Therefore, we expect to select more SNPs annotated as functionally active in ChromHMM models. To further evaluate the functional potential of the SNPs selected by T-GEN, we considered the CADD scores of the identified SNPs across all five methods (**S2 Table**). T-GEN-identified eQTLs have higher CADD scores (3.36 on average, representing a 0.9% increase compared to elnt and vb, p<2e-16, Wilcoxon rank sum test) and a higher percentage (0.34% **Fig 2B**) of functionally deleterious SNPs (larger than 20, representing a 2.3% and 2.9% increase to elnt and vb respectively). The net CADD score improvement is not substantial, which may be partially explained by the purifying selections undergone by cis-eQTLs [35,36] and their consequent low deleteriousness. These results indicate the statistically significant higher functional potential of identified eQTLs by T-GEN than those identified by other models except direct SNP filtering in elnt.annot and vb.annot, which is expected.

## More genes are effectively imputed by T-GEN

The number of genes that are assessed in transcriptome-wide association analysis is affected by the quality of expression imputation. For all five imputation methods, gene expression imputation models were filtered based on the significance of imputation models with an FDR cutoff of 0.05 (**Methods**). Therefore, significant trait-gene associations can only be detected for genes with significant imputation models.

With the same FDR cutoff, T-GEN had an increase in the range of 2.8% to 55.3% (3,807 in whole blood) for the number of gene expression imputation models in each tissue, compared with the elastic net methods in 25 of the 26 tissues (**Fig 2C and 2D**). There was a smaller increase or even slight decrease in some tissues with a smaller sample size like brain cortex (2.8%, 269, n = 136) and brain frontal cortex BA9 (-0.55%, -50, n = 118). It is worth to note that, by direct SNP filtering using epigenetic annotations, both "elnt.annot" and "vb.annot" had many fewer genes with high quality imputation models, implying that stringent SNP categorization might lead to loss in power of detecting genes with considerable expression components explained by cis-eQTL. Compared to elastic net methods (elnt and elnt.annot), variational Bayes (both vb and T-GEN) methods better imputed genes, which are partly attributed to its improved variable selection performance [37].

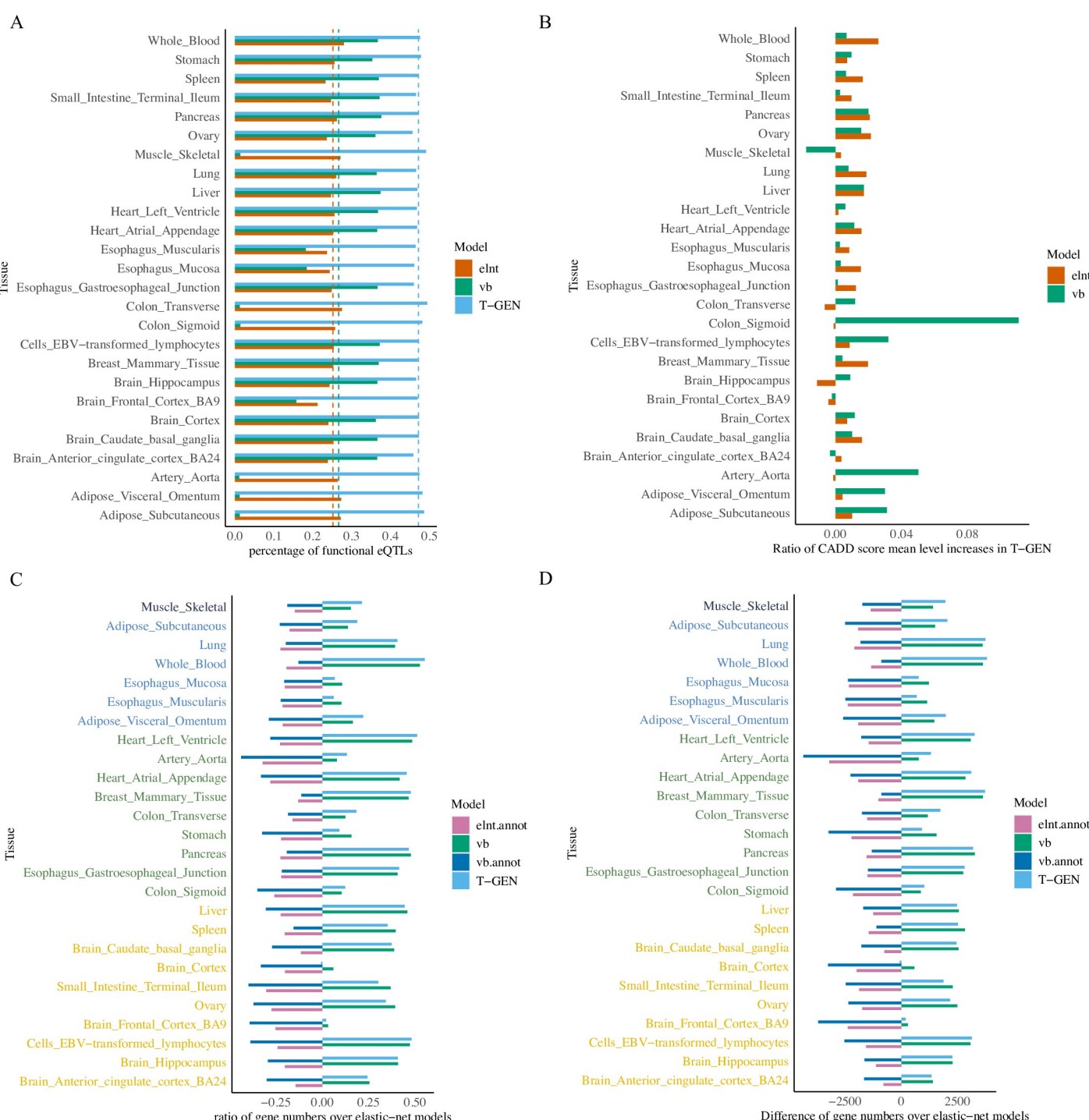

**Fig 2. More SNPs with functional potential were idenfied and more imputation models were built by T-GEN.** A) compares the percentages of SNPs in gene expression imputation models having active ChromHMM15 annotated states across three different methods (elnt, vb and vb.logit) using gene expression and genotype data from GTEx in 26 tissues. The "elnt" model was built via elastic net, the "vb" model was built via a variational Bayesian method, and the "T-GEN" model was built using our method (variational bayesian method with a logit link). Across all 26 tissues, imputation models built by our method have higher percentage of SNPs with active ChromHMM annotated states (indicated by blue bars). X axis denotes the mean percentage of SNPs in imputation models having ChromHMM15 annotated states in each tissue for each models. The dotted lines are the mean values of $R^2$ across 26 tissues for each method. B) shows the ratio of CADD score mean level increases in T-GEN compared to elnt and vb models in 26 tissues. C) shows the ratios of gene model numbers (FDR < 0.05) in each method over that in elastic net models. D) indicates the difference in the number of genes models between that from each method and that from elastic net. In C and D, different colors of each tissue indicate their sample sizes, from upper to lower: [401,501], [301, 401), [201, 301), [101, 201).

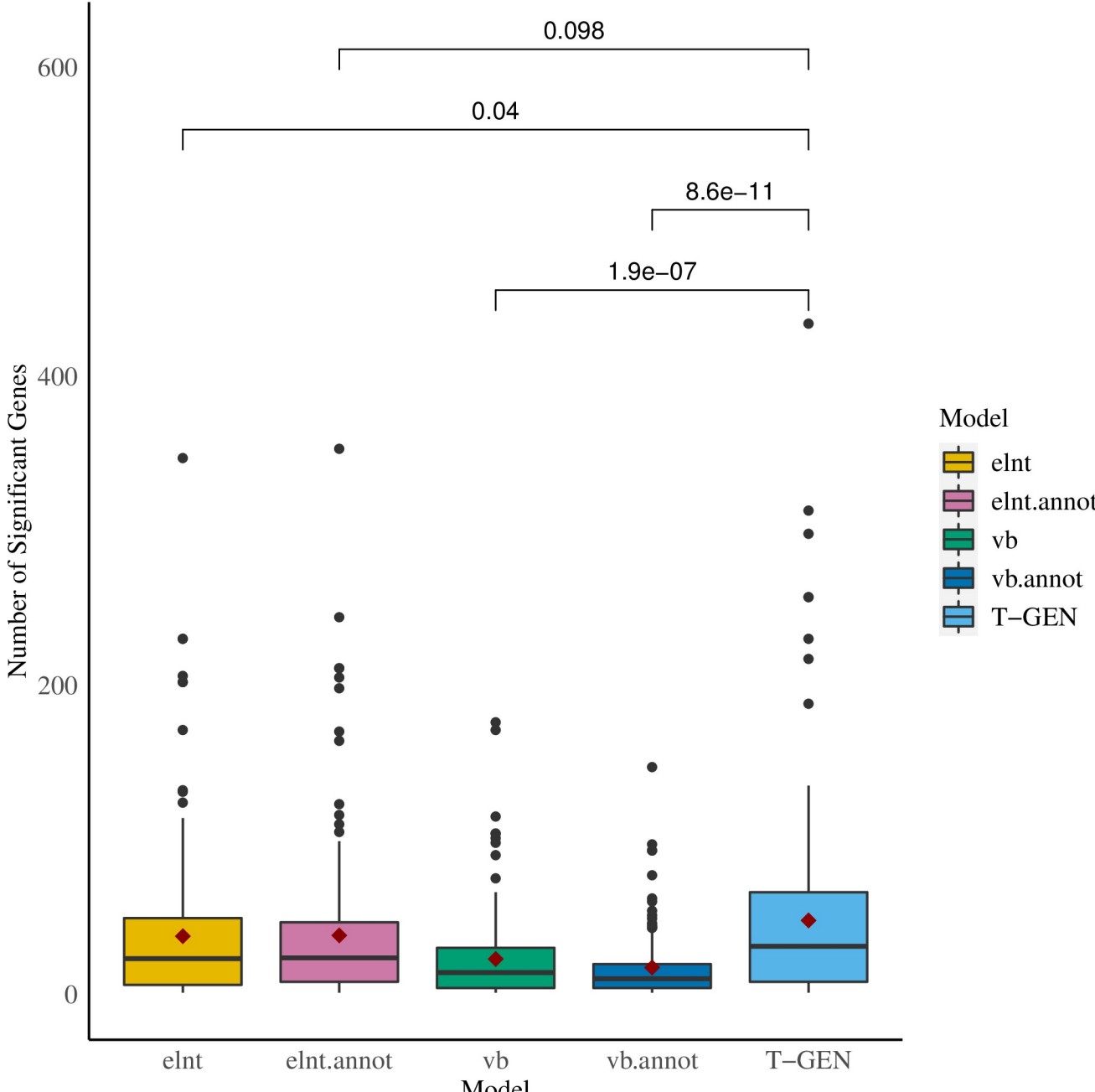

**Fig 3. More genes were identified as trait-associated by T-GEN across 207 traits from the LD Hub.** Applied to 207 traits from LD Hub, significant trait-associated genes were identified in 26 tissues (p-values threshold: 0.05 divided by the number of gene-tissue pairs). Each boxplot represents the distribution of the number of differences between that identified from our tissue-specific analysis and that identified from the four other methods.

## More genes and higher percentage of functionally conserved genes identified in 207 traits

To evaluate the performance in identifying trait-gene associations, which is the ultimate goal of gene expression imputation, we applied T-GEN to GWAS summary statistics of 207 traits (**S3 Table**) from LD Hub. After Bonferroni adjustment, significant trait-associated genes were identified in each tissue. T-GEN showed a 25% (9.6 genes, compared with elnt.annot) to 175%

(30.4 genes, compared with vb.annot) increase in the average number of significant trait-associated genes across 207 traits (**Fig 3**). After aggregating identified genes into pre-defined cyto-genetic bands, we observed a similar pattern in the numbers of identified trait-associated loci across 207 traits. T-GEN identified the largest number of associated loci (14.3 loci on average) across all five methods (**S2 Fig**), showing an 11% (1 locus, compared with elnt) to 86% increase (5 loci, compared with vb.annot).

To investigate the functional potential of trait-associated genes identified by each method, we further compared the enrichment pattern of associated genes having pLI scores larger than 0.99 for each method. Among all identified genes, some genes are identified as trait-associated in multiple traits. Grouping identified genes into three categories based on the number of associated traits, we found that T-GEN identified the largest percentage of trait-associated genes having higher pLI (>0.99) in gene groups with fewer than 5 associated traits (**Fig 4A**). This may indicate that T-GEN is more likely to identify conserved genes specific to 1–2 diseases compared to other methods (p = 0.009 to elnt, p = 0.002 to vb). Across all categories, T-GEN also showed the strongest enrichment signal (fold change: 1.15, binomial test p value: 0.038) compared to all the other four methods, which didn't show significant enrichment pattern (**S4 Table**). Although pLI score is an indirect measure of gene importance in human traits and more associated with fitness, it does provide an important way to prioritize genes whose heterozygous mutations are phenotypically harmful [38,39].

To assess the ability of identifying associated genes in the tissue most relevant to traits, which is defined as the tissue with the highest heritability enrichment estimated by LDSC [40], we compared the numbers of identified genes in the most relevant tissues (**S5 Table**). Using

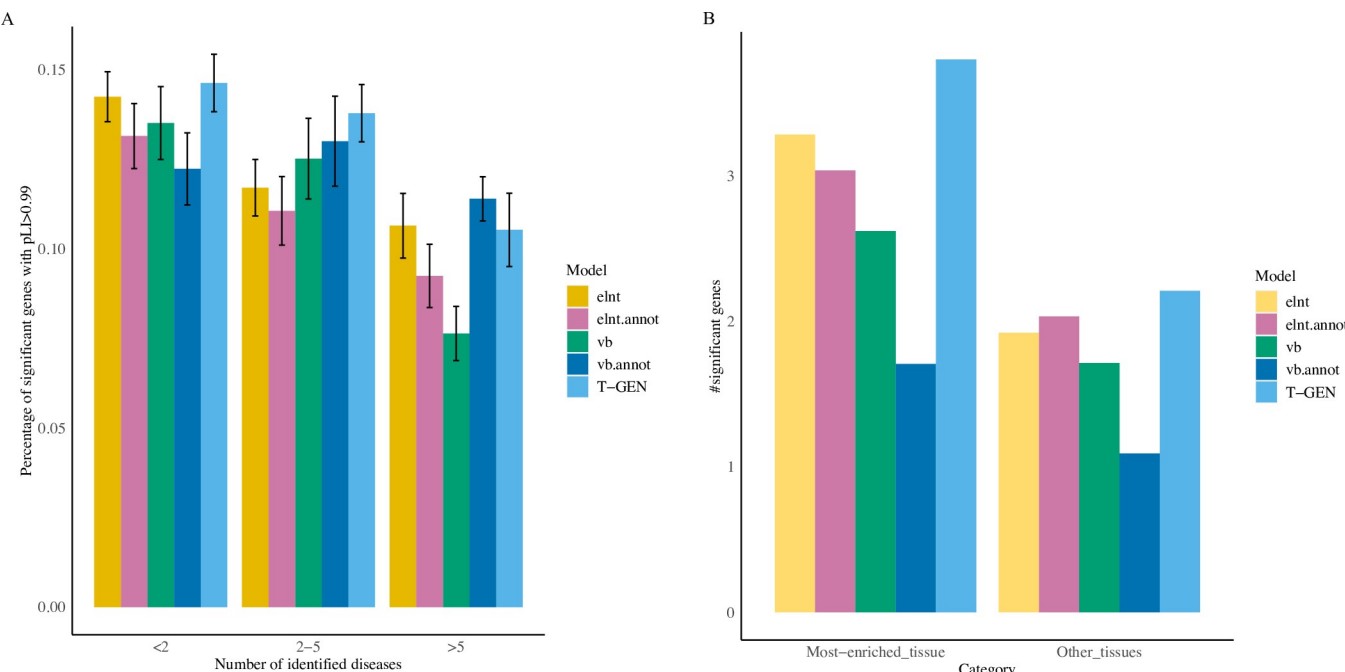

**Fig 4. Function constraint and tissue-specificity of identified trait-associated genes.** A) Higher pecentage of significant genes by T-GEN have pLI scores larger than 0.99 for genes identified in leass than 5 traits. Considering the number of traits that each identified gene is associated with, all significant trait-associated genes were grouped into three categories. The bar plot shows the percentage of genes identified by each method having larger pLI scores (>0.99) in each category. Error bars indicate the standard error calculated using bootstraping (120 traits each time, for 20 times). B) More genes were identified by T-GEN as trait-associated in tissues most enriched for genetics signals. In tissues with the highest heritability enrichment and also other tissues, the numbers of identified trait-associated genes were compared across all five methods. Each barplot shows the mean value of the numbers of identified trait-associated genes across 207 traits in the LD Hub.

LDSC and annotation from GenoSkyline-Plus [41], we identified the tissue with the highest heritability enrichment for each trait. We compared the number of significantly associated genes identified in heritability-enriched tissues across five methods (Fig 4B). In the most-enriched tissues, T-GEN identified the largest numbers of significantly associated genes. When comparing the ratios of the number of genes identified in the most-enriched tissue and those in the other tissues across 151 lipid-associated traits (S3 Fig), T-GEN showed a significant increase compared to the vb (Wilcoxon rank sum test, p = 0.048) and vb.annot models (p = 3.5e-3).These results suggest that T-GEN can better identify disease genes in trait-relevant tissues.

Overall, T-GEN showed improvement not only in the total number of trait-gene associations, but also in identifying genes with functional importance potential and tissue-specific associated genes in tissues most relevant to a trait.

## T-GEN identifies novel genes for late-onset AD

To further investigate the performance of our method in identifying trait-associated genes in detail, we analyzed the biological functions of genes associated with AD (N = 74,046) that were only identified by our method. Considering the total number of tissue-gene pairs (258,039), 96 significantly associated genes at 15 loci (S6 Table, Fig 5) were identified by T-GEN, with five loci identified in the brain tissues (caudate basal ganglia, anterior cingulate cortex BA24, hippocampus, cortex, and frontal cortex BA9) and four loci identified in the whole blood. Thirteen out of the 15 loci have been implicated in AD GWAS[42]. In comparison (S4 Fig), 79 genes at 10 loci were identified by elastic net models, 81 genes at eight loci by elnt.annot models, 61 genes at three loci by vb.annot models, and 81 genes at 11 loci by vb method. Not only the number of associated genes by T-GEN is the largest, the heritability enrichment in signal-contributing eQTLs for associated genes by T-GEN is also the highest (S7 Table).

Compared with AD-associated genes identified by the other four methods, three loci were only identified by our method. One locus is located on 14q32.12 including *LGMN* (p = 9.45e-8). *LGMN* is located about 200kb to previously identified GWAS significant SNP *rs1049863* [42]. *SLC24A4* and *RIN3* are two potential genes contributing to the GWAS signal of this locus in previous GWAS studies, whose functions are not yet understood. *LGMN* encodes protein AEP that cleaves inhibitor 2 of PP2A and may trigger tau pathology in AD brain [43], which makes *LGMN* a potential signal gene at this locus. Another locus is 16p22.3, where *COG4* was identified (p = 1.35e-7). Two of the identified eQTLs (S8 Table) of COG4 are potential GWAS hits (rs7196032, p = 1.1e-4 and rs7192890: p = 3.5e-3) (S5 Fig). *COG4* encodes a protein involved in the structure and function of the Golgi apparatus. Recent research has shown that defects in the Golgi complex are associated with AD and Parkinson's disease by affecting the functions of Rab-GTPase and SNAREs [44]. The third locus is 6p12.3, where *CD2AP* (p = 5.70e-8) and *RP11-385F7.1* (p = 1.27e-7) were identified. This locus was previously identified as a susceptibility locus in AD [42] with *CD2AP* reported as the locus signal genes affecting amyloid precursor protein (APP) metabolism and production of amyloid-beta [45]. While the additional *RP11-385F7.1* identified by our method is a long non-coding RNA (lncRNA) located near the promoter region of *CD2AP*. The level of RP11-385F7.1 lncRNA has a strong Pearson correlation with gene expression level of *COQ4* located on 16q22.3 [46], which was also the gene only identified by our method and encodes a protein that may serve as an antioxidant strategy target for AD [47].

Although other loci identified by our method have been implicated in GWAS, including *CLU* locus (8p21.2), *BIN1* locus (2q14.3), *CR1* locus (1q32.2), *MS4A6A* locus (11q12.1), *PTK2B* locus (8p21.1, 8p21.2), and *CELF1* locus (11p11.2), we identified additional, potentially

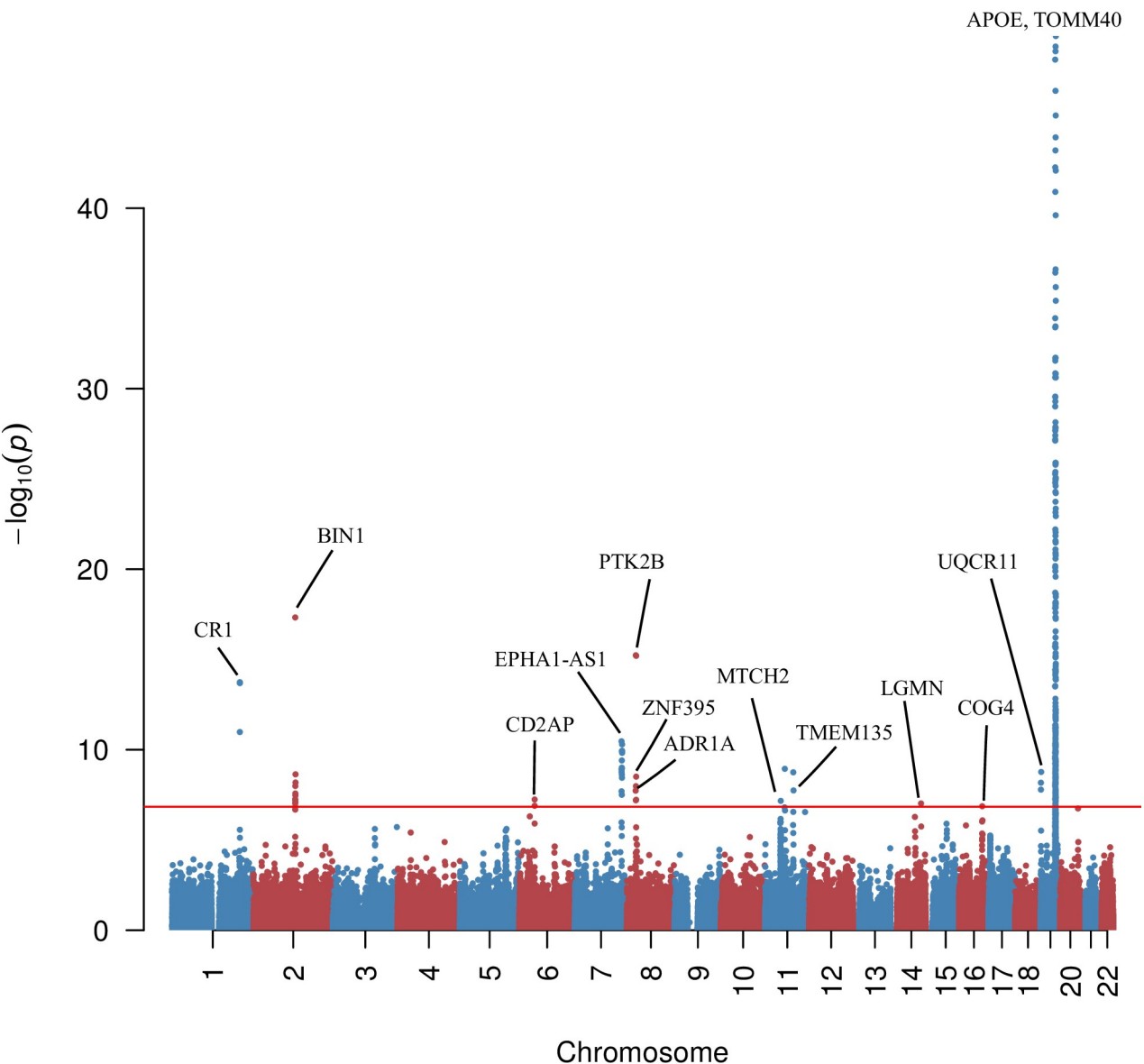

**Fig 5. Gene-level manhattan plot of T-GEN results in IGAP data.** The plot shows the gene-level association with LOAD attained from T-GEN. Several significant genes are indicated in the figure.

functionally-impactful genes at these loci. At the *PTK2B* locus, we identified an additional gene *ADRA1A* (p = 1.85e-8), which is involved in neuroactive ligand-receptor interaction and calcium signaling and has been implicated as a potential gene in late-onset AD via gene-gene interaction analysis [48]. At the *MS4A6A* locus, we identified an additional gene named *OSBP* (p = 1.56e-7), which transports sterols to nucleus where the sterol would down-regulate genes for LDL receptor, which is important in AD etiology [49].

Two out of 15 loci identified by our method are novel compared with published GWAS loci [42,50–52]. One locus is located on 16q22.3, where *COG4* was identified (p = 1.35e-7), which is also the locus only identified by our method and is discussed above. The associated gene identified at the other novel locus, *TMEM135* (p = 1.80e-8) is located 1MB (**Fig 6**) downstream of the identified GWAS locus (the *PICALM* locus), and two of the identified eQTLs of

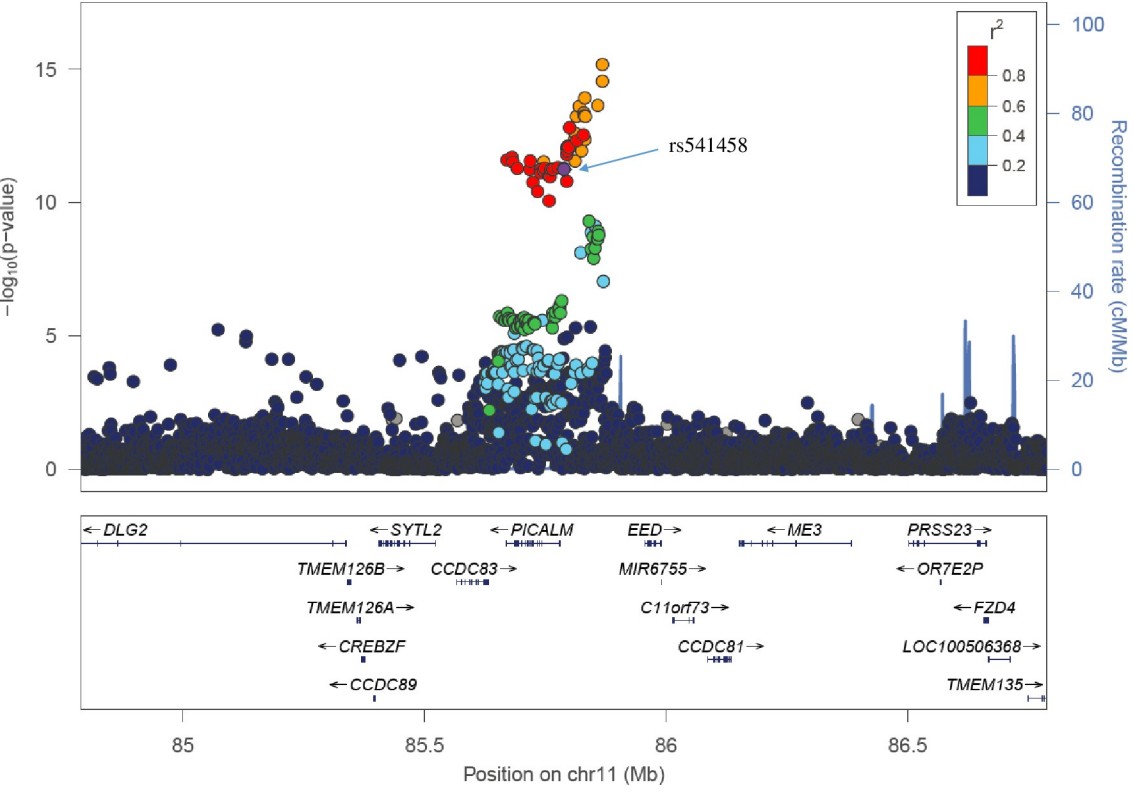

**Fig 6. Regional Manhattan plot around *TMEM135*.** The listed SNP (rs541458) is one of the identifed eQTLs by T-GEN in the imputation model of *TMEM135*. Among all eQTL of *TMEM135* identified by T-GEN, this SNP also has the strongest GWAS signal in the published AD GWAS study.

*TMEM135* by our method were located at the GWAS loci (rs536841: p = 5.27e-14 and rs541458: p = 5.67e-12). On the other hand, *TMEM135* is a target gene for liver X receptors (LXR), which is involved in removing excessive cholesterol from peripheral tissues [53]. Increased cholesterol levels in brain tissue are associated with accumulated AβPP [51] suggesting the potential role of *LXR* and *TMEM135* in the etiology of AD.

To validate the T-GEN results, we tried to replicate our findings in an independent GWAS for AD using inferred phenotypes based on family history (GWAX, N = 114,564) [54]. Among the 96 genes identified by our method using the GWAS of clinical AD, 50 (52%) genes were successfully replicated in this independent GWAS study (p < 5.2e-4) including *TMEM135* in the novel loci. For the other four methods, only 9.9% (8/81, vb) to 41% (33/81, elnt.annot) genes were replicated in the GWAX data. Pathway enrichment analysis via Enrichr [55] identified apoptosis-related network (p = 2e-3), statin pathway (p = 9e-3), and ApoE-related inflammation and atherosclerosis (p = 0.04). Statin pathway was again implicated (p = 0.067) in the GWAX data despite a lack of statistical significance. The high replication rate of the results in an external study further confirmed the power of T-GEN.

To further evaluate the impact of sample size on training gene expression imputation models, we built imputation models in AD heritability enriched tissues estimated by LDSC (liver and whole blood) using the GTEx v8 data and further identified AD-associated genes in these two tissues (**S9 Table**). Compared with using the GTEx v6 data, elastic net models by prediX-can (16 in v8 vs. 15 in v6) and T-GEN models (22 in v8 vs. 28 in v6) did not show substantial increase in the number of genes identified. However, with increased sample size in GTEx v8,

we observed higher replication rates for the identified disease-associated genes using the GWAX data. For elastic net models, the replication rate increased from 17.6% to 66.7% while for T-GEN models, the rate increased from 20.7% to 75% in these two tissues.

We also compared the performance of T-GEN models trained in GTEx v8 with a recent method named mash [56]. Compared with the most updated models built on GTEx v8 for prediXcan (trained using elastic net) and mash, T-GEN identified the largest number of genes (22 vs. 18 by mash and 16 by prediXcan) and maintained a high replication rate in the GWAX data (75% vs. 76.2% in mash and 66.7% in prediXcan).

## Discussion

In this paper, we have introduced a new method called T-GEN, which leverages epigenetic signals to improve gene expression imputation and identify trait-associated genes. Different from previous methods, T-GEN uses data from GTEx and Roadmap Epigenomics Project to prioritize SNPs with active epigenetic annotations for gene expression imputation. We found that T-GEN models were more likely to include SNPs with functional potential and more genes were effectively imputed compared to other methods. Applied to more than 200 traits, T-GEN identified more genes/loci associated with these traits. T-GEN is especially more likely to identify genes with potential function importance (high pLI scores).

When applied to AD GWAS, T-GEN identified the largest number of associated genes (96 genes in 15 loci) compared with four other methods. We found novel association signals at previously identified loci including *LGMN* in the *SLC24A4/RIN3* locus, *ADRA1A* at the *PTK2B* locus and *OSBP* at the *MS4A6A* locus. Besides, two novel loci were identified on chromosomes 3 and 16. Genes identified by our method suggest putative roles of several biological processes in Alzheimer's disease, including mitochondrial dysfunction (indicated by *UQCR11*, *MTCH2* and *TMEM135*), cholesterol transportation (indicated by *TMEM135* and *OSBP*), and neuron activity (indicated by *ADRA1A)*. Fifty out of 96 identified associated genes were replicated in an independent GWAS dataset. Most importantly, the genes identified from our method for the Alzheimer's disease showed higher replication rates than those identified from other methods.

Overall, T-GEN showed improved performance in prioritizing functional SNPs and identifying disease-associated genes. Limited by the lack of individual-level epigenetic data and small sample sizes, T-GEN may also be further improved in different aspects, such as the low accuracy of predicting gene expression levels, which may be further improved, for example by simultaneously using non-parametric Bayesian model [57] and considering epigenetic annotation. Nevertheless, the relationship between imputation accuracy and the power of TWAS-like methods, and how GWAS signals and eQTL signals contribute to final results are rather complex and worth further investigation (**S1 Text**). The eQTLs used for expression imputation in these methods also need further experimental validations. Utilizing epigenetic annotations in the reference panel from Roadmap also limits the potential of leveraging epigenetic information in gene expression imputation process, since the epigenetic signals may vary across individuals just like gene expression levels. Another limitation of our method is that we assumed SNPs with known active epigenetic signals are more likely to regulate gene expression while the regulatory effects of epigenetics are rather complex. For example, H3K9ac might be present in both actively transcribed regions and bivalent regions [58]. It has also been shown that Cytosine-phosphate diester-guanine (CpG) islands associated with gene expression may have intermediate instead of low DNA methylation levels [59]. Although trait heritability and GWAS risk SNPs are both enriched in genomic regions with active epigenetic signals in related cell types or tissues [60–64], most of these studies are based on in-silico results. Using more

data on epigenetic regulation collected from biological experiments instead of in-silico predictions may further improve the power of our approach. In addition, only cis-SNPs are used here for gene expression imputation, while a larger proportion of gene expression may be explained by trans-eQTL [65–67]. Integrating trans-eQTL into gene expression imputations would help to identify more trait-associated genes and also co-regulatory peripheral genes and core genes [68]. Methods like T-GEN, using gene expression as a mediated trait to study genetic effects and to identify disease genes, can only provide evidence for associations between genes and traits/diseases. The causal relations between identified genes and traits should be further validated by functional analysis in future research. Also, results from TWAS studies are hard to validate in silico and a benchmarking dataset would help in the comparison of different TWAS-like methods. Considering multiple TWAS methods when identifying disease genes may aid in controlling false discoveries.

By using individual-level data with larger sample sizes would further improve power and replication rates of identifying disease-associated genes under our modeling framework even though the improvement in imputing gene expression is not significant (**S1 Text**). Apart from functional analysis, gene-level fine mapping or Mendelian randomization (MR) would also help in discriminating causal relations from associations between genes and traits.

## Methods

### Bayesian variable selection model

To model the genetic effects of SNPs on the expression level of a gene in a single tissue, we used the following bi-level variable selection model to select SNPs:

$$Y = X\beta + \epsilon,$$

$$\epsilon \sim N(0, \sigma^2 I),$$

$$\beta_k | \gamma_k = 1 \sim N(0, \sigma_\beta^2 \sigma^2 I),$$

$$\beta_k | \gamma_k = 0 \sim \delta_0,$$

$$\gamma_k \sim Bernoulli(\pi_k),$$

where Y is the n×1 gene expression level from the GTEx [69] database (v6p) (n denotes the sample size in this tissue), X denotes the n×p centered genotype matrix for cis-SNPs for this gene, n is the sample size, and p is the number of SNPs. In analysis, Y denotes the gene expression in a specific tissue while X denotes the genotype matrix for SNPs located within 1MB from the upstream or downstream of the gene. We use the p×1 vector to represent the effects of those cis-SNPs on the gene expression level while $\epsilon$ is the n×1 random error vector, which is assumed to follow a normal distribution. In a typical data set with both genotype data and RNA-seq data for the same group of individuals, the sample size *n* is usually much smaller than the number of cis-SNPs for a gene (p), some constraints are needed to allow for accurate SNP selection and effect size estimation. Therefore, to select effective SNPs from thousands of candidate cis-SNPs, we assume that the effect size vector comes from a mixture of normal distribution and a point mass at zero. $\gamma_k$ is the indicator variable denoting whether the kth SNP has an effect on gene expression level. When a SNP *k* has an effect on gene expression level (i.e. $\gamma_k = 1$), we assume that its effect size follows a normal distribution with 0 mean and variance $\sigma_\beta^2 \sigma^2$.

To more accurately identify eQTL with potential biological functions, we integrate the epigenetic annotation to prioritize SNPs with active epigenetic signals including H3K4me1, H3K4me3, and H3K9ac. We only consider epigenetic signals for SNPs which have significant epigenetic markers ($p < 1e-2$). For other SNPs that don't have significant epigenetic markers, we set their corresponding annotation to be 0. To achieve this goal, we used a logit link to associate the epigenetic annotation with the probability of a SNP being an eQTL:

$$\text{logit}(\pi_k) = A_k \omega,$$

$$\omega \sim N(0, \eta^{-1} I),$$

where $A_k$ is the epigenetic signal $1 \times m$ vector for the SNP $k$ and the $m \times 1$ vector is the epigenetic signal effect vector where $m$ is the number of epigenetic signals considered, such as DNA methylation and histone marker status. We assume that the variance of the coefficients is $\eta^{-1}$.

## Prior assumptions on hyper parameters

We make the following assumptions on the hyper parameters in the model above:

$$\sigma^2 \sim IG(a, b),$$

$$\sigma_\beta^2 \sim IG(c, d),$$

$$\eta \sim \text{Gamma}(a_0, b_0),$$

i.e. we assume that these three parameters follow either inverse gamma or gamma distributions.

## Variational Bayes inference

For the convenience of defining selected SNPs, we firstly introduce PPS(k) to denote the posterior probability of a SNP $k$ being selected:

$$\text{PPS}(k) = \sum_\theta P(\gamma_k = 1 | X, Y, \theta) P(\theta | X, Y)$$

which is basically the weighted average of the posterior probability of this SNP being effective while the weight is the model likelihood, similar to that used in varbvs.

To address the computational challenge in fitting our model to high-dimensional genotype data, we applied the variational Bayes method, which is an alternative of the Markov Chain Monte Carlo (MCMC) method for statistical inference with less computation time and lower computation burden. Variational Bayes provides an analytical approximation of the posterior probability by selecting one from a family of distributions with the minimum Kullback-Leibler divergence to the exact posterior.

Under each set of chosen prior parameters $\theta = \{\sigma^2, \sigma_\beta^2, \eta\}$, variational Bayes is applied to update other parameters to get the optimal model. Besides, for a set of chosen priors, we define the posterior probability of a SNP being selected as:

$$\alpha_k = P(\gamma_k = 1 | X, Y, \theta).$$

When updating one parameter, we need to get the approximation $(Q^*(.))$ of its posterior distribution via taking the expectations of all the other parameters in the full probability

equation:

$$P(Y|X, A, \theta) = \int \int \int P(Y|\beta, \epsilon, X)P(\sigma_\beta^2, \sigma^2, \gamma)P(\gamma|\pi)P(\pi|A, \omega)P(\omega|\eta)\mathrm{d}\eta\mathrm{d}\sigma^2 d\sigma_\beta^2.$$

More details about fitting the model were shown in the **S1 Text**.

## Model training and evaluation

Gene expression level imputation models for 26 tissues were trained using the RNA-seq and genotype data from the GTEx(v6p) project and epigenetic data from the Roadmap Epigenomics Project. Only tissues with both epigenomics data from the Roadmap Epigenomics Project and the GTEx data were considered. For genotype data, we first removed less common SNPs (minor allele frequency < 0.01) and SNPs with ambiguous alleles. For each gene, we considered SNPs located from 1 Mb upstream of its transcription starting site to 1 Mb downstream of its transcription end site. For the RNA-seq data from GTEx, we first normalized these data and further adjusted for possible confounding factors including sequencing platform, top three principal components, sex and probabilistic estimating of expression residuals (PEER) factors. More specific details of preprocessing expression data from GTEx were described in our previous publication [27].

We further used five-fold cross validation to evaluate our gene expression imputation models. More specifically, for each tissue, samples were randomly divided into five groups with about the same size. We compared our method with four other methods via training models using four groups of data and then testing them on the fifth group. To compare performance among different methods, squared correlation between the observed and imputed expression levels ($R^2$) was used. In our method, with the number of epigenetic categories existing in Roadmap varying across different tissues (**S10 Table**), all available epigenetic annotation categories were used for each tissue. The continuous annotation values (fold enrichments compared to expected background counts for ChIP-seq or DNase signal and fractions of methylation reads for DNA methylation) were used for training models. For elastic net models (elnt), as what previous studies did [7], the parameter $\alpha$ was set to be 0.5 as in PrediXcan and the optimal $\lambda$ was selected via the function cv.glmnet provided in the 'glmnet' package [70]. For elastic net models with epigenetic signals direct filtering (elnt.annot), we first removed cis-SNPs without positive values for H3K4me1 [71], H3K4me3 [72] or H3K9ac [73] signals, which have been reported as associated with gene expression regulation. After filtering, the remaining cis-SNPs were used to build imputation models. As for methods directly using variational Bayes (vb), the model is similar to our method, without the logit link of epigenetic annotations. We also applied the direct variational Bayes method to SNPs with some epigenetic signals (same filtering process in elnt.annot). Paired Wilcoxon signed rank test was used to compare the performance of the models across different genes in each tissue. For further validation, different models were used to predict gene expression levels in an external brain expression data set of the CommonMind Consortium (www.synapse.org/CMC). For gene-trait association identification, only gene expression imputation models with significant squared correlation between the observed and imputed expression levels ($R^2$) (FDR < 0.05) were considered.

## Association analysis

After estimating SNP effects on gene expression levels, we used published GWAS summary-level results to identify the gene-level genetic effects on different traits mediated by gene expression. Just as the imputation model introduced above, for a gene, in a single tissue, its gene expression is modeled via the genotypes of cis-SNPs $Y = X\beta+\epsilon$ while the expression-trait

relation is modelled by $T = \mu + Y\kappa_Y + \tau$, where $\mu$ is the intercept and $\tau$ is the error term in the model. The basic idea of this test is similar to what is used in UTMOST and PrediXcan.

The test statistic for the effect of gene expression Y on trait T is

$$Z = \frac{\widehat{\kappa}_Y}{se(\widehat{\kappa}_Y)}$$

where $\widehat{\kappa}_Y$ is the estimated expression effect on the trait and $se(\widehat{\kappa}_Y)$ stands for the stand error for the estimated effect. For a linear model, $\widehat{\kappa}$ has the following form

$$\widehat{\kappa_Y} = \frac{cov(Y, T)}{var(Y)} = \frac{\widehat{\beta}^T cov(X, T)}{var(Y)} = \frac{\widehat{\beta}^T var(X)\widehat{B}}{var(Y)}$$

where $\widehat{\beta}$ is the vector of estimated SNP effects on gene expression levels, $\widehat{B}$ denotes the vector of GWAS SNP-level effect sizes for those identified effective SNPs in the gene expression imputation model. Besides, var(X) is a diagonal matrix whose elements are the genotype variances for each SNP and var(Y) is the variance of the imputed expression levels for this gene in the tissue.

To calculate the test statistic z score, we also need to derive the stand error of the estimated $\kappa_Y$:

$$se(\widehat{\kappa_Y}) = \sqrt{var(\widehat{\kappa_Y})} = \sqrt{\frac{var(\tau)}{var(Y) \times n_{GWAS}}}$$

Besides, based on the three linear models between any two out of these three variables (genotypes, imputed expression $Y$, and trait phenotypes $T$), we can get the variance proportions of outcomes explained by predictors as:

$$R^2_{T,Y} = \widehat{\kappa}^2_Y \frac{\widehat{\sigma}^2_Y}{\widehat{\sigma}^2_T}$$

$$R^2_{T,X} = \widehat{\kappa}^2_X \frac{\widehat{\sigma}^2_X}{\widehat{\sigma}^2_T}$$

where $\widehat{\kappa}_X$ is the estimated SNP-level effect size presented in the GWAS result and $\widehat{\sigma}^2_X$ is the estimated SNP variance diagonal matrix. Therefore, the standard error of $\widehat{\kappa}_Y$ is

$$\sqrt{\frac{se(\widehat{\kappa}_X)\widehat{\sigma}^2_X(1 - R^2_{T,Y})}{(1 - R^2_{T,X})\widehat{\sigma}^2_Y}}.$$

Then, the Z score is $Z \approx \widehat{\beta}^T \Lambda \widehat{Z}_X$, where $\widehat{\beta}$ is the SNP coefficient vector in the gene expression imputation model, $\Lambda$ is a diagonal matrix whose diagonal elements are the ratio of standard errors of SNPs over the standard error of imputed gene expression. Besides, the $\widehat{Z}_X$ is the Z score vector provided in the GWAS results. Intuitively, the statistic is the weighted sum of GWAS z-scores for SNPs selected in gene expression imputation models and weights are proportional to the variance proportion of imputed expression levels explained by each SNP.

## GWAS data analysis

The summary statistics files of 207 traits (not based on UK biobank data) were downloaded from the LD Hub website (http://ldsc.broadinstitute.org/gwashare/) by March of 2018, imputation models built by all five methods across 26 tissues were applied to identify trait-associated

genes. For these five methods, individual-level genotypes from the GTEx database were used to calculate the standard deviations of SNPs and also the stand errors of imputed gene expression.

## Supporting information

**S1 Fig. Number of identified SNPs by each method.** The figure shows the number of identified eQTLs by all five methods across 26 tissues.
(PDF)

**S2 Fig. More loci were identified as trait-associated by T-GEN across 207 traits from the LD Hub.** Applied to 207 traits from the LD Hub, significant trait-associated genes were identified in 26 tissues (p-values threshold: 0.05 divided by the number of gene-tissue pairs). Those identified associated-genes were further grouped into pre-defined cytobands. Each boxplot represents the distribution of the number differences between those identified from our tissue-specific analysis and those identified from the four other methods. Y axis is truncated at the value of 3 times the third quarters of each boxplot for visualization.
(PDF)

**S3 Fig. The ratio of number of genes identified in trait-relevant tissues over that identified in non-associated tissues.** Each boxplot shows the distribution of the ratios across 207 traits in the LD Hub.
(PDF)

**S4 Fig. The Venn diagram of genes identified as associated with LOAD in all five methods.** Twenty nine genes were identified by all five methods. 62 genes were shared between T-GEN and elastic-net methods, which may indicate the consistency of gene findings in T-GEN and other TWAS methods based on elastic-net models.
(PDF)

**S5 Fig. Regional Manhanttan plot for SNPs near the identified associated-gene *COG4*.** The listed SNP (rs7196032) is one of eQTLs identified by T-GEN in the imputation model of *COG4*.
(PDF)

**S6 Fig. Evaluating the imputation accuracy of the five methods.** a) indicates the comparison of $R^2$ between observed and imputed gene expression levels in 5-fold cross validation analysis. Using 5-fold cross validation in the GTEx data, R2 between imputed and observed expression levels was calculated in elastic net (elnt) and vb.logit models. Dotted lines indicate the mean values of R2 for each method across all tissues. The mean values of vb.annot and T-GEN models are very close to each other, which lead to the overlaid green and blue dotted lines. b) shows $R^2$ between observed gene expression in the CommonMind dataset and predicted gene expression based on five different methods. T-GEN showed the lowest values of $R^2$ among all five methods. Y axis is truncated at the value of 2 times the third quarters of each boxplot for visualization. c) Using GTEx v8 data, T-GEN, elnt and mashr models were trained in the brain cortex BA9 tissue. $R^2$ between observed gene expression in the CommonMind dataset and predicted gene expression were compared across these three methods.
(PDF)

**S7 Fig. The percentages of identified trait-associated genes in 207 traits having pLI$>$0.99 and their relationship with their imputation accuracy.** For each method, all trait-associated gene-tissue models are classified into 20 bins based on their imputation accuracy ($R^2$). For each bin, the percentage of genes having pLI$>$0.99 is calculated. The linear equation indicates

the linear relationship between the mean $R^2$ and the percentage and the p value indicate the significance level of the association.
(PDF)

**S8 Fig. The percentage of genes identified as trait-associated in any of 207 traits and its relationship with the corresponding imputation accuracy.** For each method, all gene-tissue models (not just trait-associated ones) are classified into 20 bins based on their imputation accuracy ($R^2$). For each bin, the percentage of gene-tissue models identified as trait-associated (bars) and the mean level of imputation accuracy for trait-associated ones (squares with crosses) are calculated. The linear equation indicates the linear relationship between the mean imputation accuracy and the percentage of trait-associated gene-tissue models. The p value indicates the significance level.
(PDF)

**S9 Fig. The imputation accuracy in muscle skeletal tissue using GTEx v8 data.** A) shows the comparision among imputation models using binary-coded annotation, models trained using the probit link function in the annotation layer and models trained using the original T-GEN method. b) shows the results of models trained using annotation information with different missing rates. The rate 0 indicates the results of the original T-GEN method. Red diamonds indicate the mean level of each group.
(PDF)

**S1 Table. Active states in ChromHMM-15 model.** This table shows 11/21 active states in the ChromHMM-15 model.
(XLSX)

**S2 Table. Ratio of identifed SNPs with CADD score lager than 20 by each methods.**
(XLSX)

**S3 Table. 207 traits from LD Hub.** 207 traits from the LD Hub considered in our paper and also their corresponding GWAS studies.
(XLSX)

**S4 Table. Enrichment pattern of genes with pLI $>$ 0.99 in identified trait-associated genes.** This table shows the enrichment analysis results for all five methods, the p values were obtained using binomial test.
(XLSX)

**S5 Table. Numbers of singifcant genes in the most relevent tissue across traits in LDhub.** This table shows the number of significant genes in the tissues with most-enriched heritability across different traits.
(XLSX)

**S6 Table. T-GEN results in LOAD case study.** This table shows T-GEN association results in LOAD, genes with green background in the table are those replicated in GWAX data.
(XLSX)

**S7 Table. Heritability enrichment results in LOAD.** eQTLs contributing to identified LOAD-associated genes were identified as an annotation used in LD score regression. Heritability enrichment analysis was further conducted. The table shows the results for all five methods.
(XLSX)

**S8 Table. eQTL for novel genes identified in AD by T-GEN.** This table shows eQTLs for novel AD-associated genes identified by T-GEN.
(XLSX)

**S9 Table. Genes identified associated with LOAD using GTEx v8 models.** This tables shows identfied LOAD-associated genes by mash, prediXcan and T-GEN in heritability-enriched tissues of LOAD (whole blood and liver). Whether the gene is replicated in the additioanl GWAX data is also indicated in the table.
(XLSX)

**S10 Table. Numbers of Roadmap annotation categories for each cell types and corresponding tissues.** This table shows the numbers of roadmap annotation categories (like DNA methylation, H3K4me3 and H3K9ac) in Roadmap cell types and corresponding 26 Roadmap tissues.
(XLSX)

**S11 Table. CPU hours needed for building gene expression imputation models in T-GEN.** This table shows the running time needed for building imputation models in T-GEN across 26 tissues using GTEx v6 dataset.
(XLSX)

**S12 Table. AD enrichment.** Heritability enrichment of AD in GenoSkyline annotations using LDSC.
(XLSX)

**S13 Table. Model number.** T-GEN models in each tissue.
(XLSX)

**S1 Text. Details of the T-GEN method, discussion on the effects of imputation accuracy and effects of the annotation layer.** In the supplementary method part, we showed the details of the variational method used in our T-GEN model including the updating procedures of parameters. In the supplementary discussion part, the influences of gene expression imputation accuracy on gene-trait association test is discussed. Also, the effects of different link function in the annotation layer of T-GEN model, ways of configuring annotation information and incompleteness of annotation were discussed.
(DOCX)

## Author Contributions

**Conceptualization:** Wei Liu, Mo Li.

**Data curation:** Wei Liu, Wenfeng Zhang, Geyu Zhou, Xing Wu, Jiawei Wang.

**Formal analysis:** Wei Liu, Geyu Zhou, Xing Wu, Jiawei Wang.

**Funding acquisition:** Hongyu Zhao.

**Investigation:** Wenfeng Zhang, Qiongshi Lu.

**Methodology:** Wei Liu, Mo Li.

**Project administration:** Hongyu Zhao.

**Resources:** Wei Liu, Geyu Zhou.

**Software:** Wei Liu.

**Supervision:** Hongyu Zhao.

**Validation:** Wei Liu.

**Writing – original draft:** Wei Liu.

**Writing – review & editing:** Wei Liu, Mo Li, Geyu Zhou, Xing Wu, Jiawei Wang, Qiongshi Lu, Hongyu Zhao.

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
