## [Decision Letter · Decision Letter 0]

3 May 2020

Dear Prof. Zhao,

Thank you very much for submitting your manuscript "Leveraging functional annotation to identify genes associated with complex diseases" for consideration at PLOS Computational Biology.

As with all papers reviewed by the journal, your manuscript was reviewed by members of the editorial board and by several independent reviewers. While the paper proposes a methodology for solving an important problem, the reviewers raised concerns regarding application of the method to analyze data. In light of the reviews (below this email), we would like to invite the resubmission of a significantly-revised version that takes into account the reviewers' comments. 

We cannot make any decision about publication until we have seen the revised manuscript and your response to the reviewers' comments. Your revised manuscript is also likely to be sent to reviewers for further evaluation.

Sincerely,

Seyoung Kim

Guest Editor

PLOS Computational Biology

Jian Ma

Deputy Editor

PLOS Computational Biology

Reviewer's Responses to Questions

**Comments to the Authors:**

Reviewer #1: Liu et al introduced a new method called T-GEN (Transcriptome-mediated identification of disease-associated Gens with Epigenetic aNnotation) to identify disease-associated genes leveraging epigenetic information. They incorporate epigenomic annotation in their formula as a Bayes prior (variational Bayes method) in imputing gene expression levels and then perform TWAS (transcriptome-wide association study) of disease traits.

This method leverages functional annotation to identify genes associated with complex diseases. They applied T-GEN to 207 complex traits and identified more trait-associated genes (ranging from 7.7 % to 102%) than those from existing methods. Among the identified genes associated with these traits, T-GEN can better identify genes with high (>0.99) pLI scores (higher scores indicate more functional importance of genes) compared to other methods. When T-GEN was applied to late-onset Alzheimer’s disease, they identified 96 genes located at 15 loci, including two novel loci not implicated in previous GWAS. I believe that the method proposed in this paper is an important extension of existing methods in identifying disease associated genes. The paper has been well written. I have the following comments.

1. The authors assumed that SNPs with active epigenetic annotations are more likely to regulate tissue-specific gene expression. I think this is only partially true given our knowledge about human genome. More GWAS SNPs are located in the non-coding and unknown regions. In addition our knowledge about epigenetic annotations is limited, mainly based on prediction by statistical models. Therefore, this assumption is clearly a limitation to the method but may not have a good solution at this point. I wonder if the authors can comment on this and provide some insights in the discussion.

2. ‘Reported GWAS hits are enriched in regions with active epigenetic signals and they can help fine-map true GWAS hits with functional impacts’. I wonder what proportion of GWAS hits have active epigenetic signals.

3. What does ‘positive value’ mean in ‘Removing cis-SNPs without positive values for H3K4me1, H3K4me3, or H3K9ac signals’? What are the effects of the SNPs that show positive values with these three genes?

4. How did the author form a continuous epigenetic annotation score? Did they compare different ways to formulate the annotation score? How did the author choose the current method to form continuous annotation? Is there an advantage to use logit link to associate the epigenetic annotation with the probability of a SNP being an eQTL?

5. The GWAS SNPs with active epigenetic annotation are prioritized. I wonder how they treated the GWAS SNPs without clear annotation.

6. Bayes methods are known to be computational intensive, especially for high-dimensional genotype data. The authors used the variational Bayes method which is less computational intense compared to MCMC. I wonder if the authors can provide the length of hours for imputation of gene expression on each tissue type. I assume that it takes longer in some tissue types and shorter in other tissue types.

Reviewer #2: Liu et al. proposed an interesting idea to incorporate epigenetic annotation into transcriptome-wide association studies (TWAS). They adopted a Bayesian variable selection model to integrate Roadmap epigenetic annotations, GTEx data, and LD hub GWAS summary statistics. Through intensive analyses of 207 traits on LD hub, they demonstrated advantages of their proposed method T-GEN over other methods. Then they thoroughly discussed a detailed application to Alzheimer’s disease. Below I list some comments that may help improve the manuscript.

1. It is worrisome that using GTEx brain cortex BA9 data to impute CommonMind Consortium (CMC) BA9 brain data can only achieve R2 of 0.007. Given GTEx data are only healthy controls, how about only considering CMC controls data with the same age range as GTEx subjects?

PrediXcan published its prediction model online (http://predictdb.org/), including for a newer version of GTEx data (V8). What is the R2 directly using their prediction weights?

It is also counterintuitive that improving gene expression imputation accuracy does not help TWAS. How many genes are left after the FDR filtering?

2. The authors claimed T-GEN discovered more genes. Is there any empirical check of the distribution of the p-values to ensure that there is no inflated FDR?

3. The GTEx data have been updated to V8. Given the largely increased sample size, it may help boost gene expression imputation accuracy and association discovery if the authors can update the analyses to GTEx V8 data. If not for all the traits, updating the main application (Alzheimer’s disease) would show the impact of sample size.

4. Fig 2 shows the percentage of functional SNPs. Can the authors show the number of selected SNPs by each method? Why two methods (elnt.annot and vb.annot) are not compared here?

The authors acknowledged that “We note that T-GEN utilizes the epigenetic information in SNP selections, and the same information is also used in ChromHMM models. Therefore, we expect to select more SNPs annotated as functionally active in ChromHMM models. To further evaluate the functional potential of the SNPs selected by T-GEN, we considered the CADD scores of the identified SNPs across all five methods.” So what presented in Fig 2 is an unfair comparison, and it seems to make more sense to demonstrate the results of CADD scores in Fig 2.

5. Fig 3 is not cited in the paper. There are 8 main figures and 8 supplementary figures. The authors may consider combining similar figures into a bigger figure or compile all supplementary materials in a single file. Now it takes time to download and review supplementary figures one by one.

6. As a computational biology paper, would the authors provide analysis code or software?

7. The authors considered 26 tissue types with Roadmap and GTEx data. For instance, for the application to Alzheimer’s disease, they identified loci in tissue types like ovary and lung. I am not sure if this makes sense in biology. The authors may pursue further on trait-relevant tissue types. E.g., 1) provide heritability enrichment for each identified tissue type; 2) weight tissue types by trait relevance in the testing; 3) only consider relevant tissue types by a cutoff of heritability enrichment. This would also help release the burden of multiple testing.

Typos:

1. “gens” appeared four times, including when mentioning the full name of the proposed method T-GEN. Should it be “genes”?

2. “relavant” in the caption of Fig S2.

3. S1 Text mentioned that there are 208 traits other than 207 stated in the main text.

Reviewer #3: The main idea of this study is to add epigenetic information in gene expression imputation from eQTL SNPs and this helps to identify more statistically predictive and biologically functional SNPs, which leads to the identification of more trait-associated genes. In the process of imputing expression with SNP information, the epigenetics annotation was used to set priorities among candidate SNPs. Experimental results show that the proposed method can select more potent factors than the previous methods.

Major comment:

The study is based on the assumption that SNPs with active epigenetic annotation are more likely to modulate tissue-specific gene expression. Reference to support this or experimental validation would be necessary (paragraph 2 in Introduction).

As the authors noted, it is naturally expected to select more SNPs annotated as functional by adding epigenetic information in the imputation process. Therefore, further validation of selected SNPs is done by showing that the T-GEN identified eQTLs have higher CADD scores, while the score improvement is marginal. The final validation is done through the identified train-associated genes, which are more in number and include higher percentage of functionally conserved genes. Still, it doesn’t seem to fully validate the potential advantage of the proposed method.

Since the main idea in Methodological perspective is to add epigenetic annotation information, it would be helpful to do some experiments regarding the robustness against the bias or incompleteness of the annotation information. Simulation study on this and other aspects that can show the performance behavior of the proposed method would be useful.

This method was compared to Elnt and vb, which are relatively classic methods. I’m wondering if there is more recent research or method to use for comparison.

In the section “More genes can be effectively imputed by T-GEN”, more detailed explanation of the measurement method is needed. In addition, there should be comparison of prediction errors as to whether the proposed method predicts expression matrix Y well from X.

Figure 3 is not explained in the main text. Also, there is an elnt category in Figure 3, and it seems that the values are all zero. If it is just a control (all 0), it may be deleted.

Typo:

In line 297, “gens” should be “genes”.

**Have all data underlying the figures and results presented in the manuscript been provided?**

Reviewer #1: Yes

Reviewer #2: None

Reviewer #3: None

PLOS authors have the option to publish the peer review history of their article (what does this mean?). If published, this will include your full peer review and any attached files.

Reviewer #1: No

Reviewer #2: No

Reviewer #3: No
---

## [Decision Letter · Decision Letter 1]

14 Aug 2020

Dear Prof. Zhao,

Thank you very much for submitting your manuscript "Leveraging functional annotation to identify genes associated with complex diseases" for consideration at PLOS Computational Biology. As with all papers reviewed by the journal, your manuscript was reviewed by members of the editorial board and by several independent reviewers. The reviewers appreciated the attention to an important topic. Based on the reviews, we are likely to accept this manuscript for publication, providing that you modify the manuscript according to the review recommendations, specifically on making the software available and proofreading the text.

Sincerely,

Seyoung Kim

Guest Editor

PLOS Computational Biology

Jian Ma

Deputy Editor

PLOS Computational Biology

[LINK]

Reviewer's Responses to Questions

**Comments to the Authors:**

**Reviewer #1:** I believe the authors have appropriately addressed my comments. I feel that the authors used many 'can' in sentences. Some of the 'can' should be changed to 'may', other 'can' should be removed and just use the verb. For example, “H3K9ac [can] be present in both actively transcribed and bivalent regions." In this sentence, 'can' should be 'might'. Another example, "More specifically, the running

time for model training in each tissue [can] be found in the S11 Table", here should be ".... in each tissue was displayed in the S11 Table". Another "the annotation configuration and

incompleteness [can] also affect the results", here "can" should be "may", "More genes [can] be effectively imputed by T-GEN", here "can be" should be changed to "are"

**Reviewer #2:** My comments have been addressed.

**Reviewer #3:** This paper proposes a new method of using functional annotation to identify disease-related genes and it seems a significant contribution to the field. Most of the concerns I have raised about the previous version have been resolved and I recommend to accept this manuscript after minor edits, e.g.

- line 336: substantial number increase in the number of genes  substantial increase in the number of genes

And the authors would need to include in the manuscript a link that provides the used data and software/code.

**Have all data underlying the figures and results presented in the manuscript been provided?**

Reviewer #1: Yes

Reviewer #2: Yes

Reviewer #3: None

PLOS authors have the option to publish the peer review history of their article (what does this mean?). If published, this will include your full peer review and any attached files.

Reviewer #1: No

Reviewer #2: No

Reviewer #3: No
---

## [Editor Report · Decision Letter 2]

5 Sep 2020

Dear Prof. Zhao,

We are pleased to inform you that your manuscript 'Leveraging functional annotation to identify genes associated with complex diseases' has been provisionally accepted for publication in PLOS Computational Biology.

Best regards,

Seyoung Kim

Guest Editor

PLOS Computational Biology

Jian Ma

Deputy Editor

PLOS Computational Biology

---

## [Editor Report · Acceptance letter]

20 Oct 2020

PCOMPBIOL-D-20-00202R2 

Leveraging functional annotation to identify genes associated with complex diseases

Dear Dr Zhao,

I am pleased to inform you that your manuscript has been formally accepted for publication in PLOS Computational Biology. Your manuscript is now with our production department and you will be notified of the publication date in due course.

With kind regards,

Laura Mallard
